# Environmental Correlates of Sexual Signaling in the Heteroptera: A Prospective Study

**DOI:** 10.3390/insects12121079

**Published:** 2021-11-30

**Authors:** Eleanor H. Z. Gourevitch, David M. Shuker

**Affiliations:** Centre for Biological Diversity, Insect Behavioural Ecology, School of Biology, University of St Andrews, Sir Harold Mitchell Building, Greenside Place, St Andrews KY16 9TH, UK; David.shuker@st-andrews.ac.uk

**Keywords:** sexual selection, Heteroptera, sexual communication, stridulation, abdominal vibration, antennation, chemical signaling

## Abstract

**Simple Summary:**

Sexual selection influences a broad range of morphological, behavioral, and physiological characteristics, helping to drive the divergence of populations, up to and including the formation of new species. However, we are still limited in our ability to predict what traits sexual selection may favor and under what circumstances. We addressed sexual selection in the Hemiptera, also known as true bugs, identifying four main forms of sexual communication used by the sub-order Heteroptera (chemical signals, acoustic signaling via stridulation, abdominal vibration, and tactile signaling via antennation). We compared how these modes of sexual communication occur within three broad habitat types in which they are found (leaf-litter, plant-based, and semi-aquatic habitats), reviewing each mode of communication, their environmental context, and providing a wide range of examples of their occurrence within the Heteroptera. We argue that looking for associations between mechanisms of sexual selection and particular ecologies will help us move towards a more predictive theory of sexual selection. In our rapidly changing world, these kinds of studies may also be important in understanding the role sexual selection will play in determining species persistence.

**Abstract:**

Sexual selection is a major evolutionary process, shaping organisms in terms of success in competition for access to mates and their gametes. The study of sexual selection has provided rich empirical and theoretical literature addressing the ecological and evolutionary causes and consequences of competition for gametes. However, there remains a bias towards individual, species-specific studies, whilst broader, cross-species comparisons looking for wider-ranging patterns in sexual selection remain uncommon. For instance, we are still some ways from understanding why particular kinds of traits tend to evolve under sexual selection, and under what circumstances. Here we consider sexual selection in the Heteroptera, a sub-order of the Hemiptera, or true bugs. The latter is the largest of the hemimetabolous insect orders, whilst the Heteroptera itself comprises some 40,000-plus described species. We focus on four key sexual signaling modes found in the Heteroptera: chemical signals, acoustic signaling via stridulation, vibrational (substrate) signaling, and finally tactile signaling (antennation). We compare how these modes vary across broad habitat types and provide a review of each type of signal. We ask how we might move towards a more predictive theory of sexual selection, that links mechanisms and targets of sexual selection to various ecologies.

## 1. Introduction

Sexual signaling is at the heart of sexual selection. Throughout the mating sequence, communication within- and between the sexes is often key to success in competition for mates and access to their gametes [1,2]. Communication within the sexes in terms of sexual selection typically involves ritualized displays of dominance, fighting ability, or territory ownership, that precede, and potentially allow individuals to avoid, actual physical contests between rivals [3,4,5,6]. In terms of between-sex communication, whilst the focus has perhaps traditionally been on courtship displays and subsequent mate choice, for many organisms, communication is also an important part of (competitive) mate searching and will include signaling up to and during copulation [7,8,9,10]. The signals used to attract and entice a mate can vary greatly in complexity and signal content, including information associated with species discrimination, as well as one or more components of mate quality, and they can also vary in methods of transmission and sensory modality (visual, auditory, tactile, or olfactory). These signals do not evolve in isolation though, and the effectiveness of signals is influenced by both the responses of the receivers [5,11,12,13,14] and the environment in which the signaling occurs.

Signals involved in sexual communication have evolved to maximize efficient signal transfer either to as many potential receivers as possible or as efficiently as possible to a focal receiver, for instance, the audience for a courtship display [5]. Therefore, selection favors signals that optimize the signal to background noise ratio, whilst minimizing signal degradation [5]. This is highly dependent on the environment in which sexual communication occurs [9,15,16,17,18,19,20]. Environmental conditions, both biotic and abiotic, spanning climate, seasonality, habitat structure, predators, and prey, interact via natural selection to influence how sexual selection acts, in terms of behavior, morphology, and key life-history allocation decisions [19]. Moreover, one might think that communication only consists of two parties, the signaler, and the receiver, but these signals are under constant threat of eavesdropping from parasites, predators, and rivals [21,22]. The way that the environment shapes sexual signals, due to the physical constraints placed on the signaler and the receiver by the environment, has been increasingly well-explored, for instance by Endler and colleagues across a variety of contexts [23,24,25]. For example, how habitat shapes light levels influences the receptivity of mates to color signaling in several species of fish [23,26,27]. In other examples, animals might seek out habitats that better enhance their sexual behaviors. Male wolf spiders drum their abdomen against the ground, a behavior used to attract mates, and these spiders prefer dry leaves as a habitat and drumming substrate [28]. Such environmental influences have been shown to cause changes in both habitat choice and competitive behaviors [17]. More generally, the extent to which visual, acoustic, or chemical signaling is deployed will be shaped in large part by the transmission characteristics of the environment, involving factors such as light levels, humidity, temperature, exposure, vegetation structure, water turbidity, and so on. Therefore, habitat-and the environment more generally–should be an important contributor to the trajectory of sexually selected characteristics and behaviors.

Sexual selection has, of course, been a key part of evolutionary biology for the last five decades, re-emerging into prominence thanks to both the rehabilitation of mate choice as an important mechanism of sexual selection [1,29,30,31], and the discovery of post-copulatory sexual selection (first as sperm competition, then as a cryptic female choice [32,33,34,35]). However, there remains a focus on both a few model species [36] and individual, species-specific studies. While such studies are crucial for answering some questions, broader comparative studies across species and habitats remain uncommon (see, for instance, discussion of both issues in [37]). As such, we are still lacking a framework that links broader patterns of sexual selection and sexually selected traits to underlying species ecologies. In particular, we currently struggle to predict what particular traits we would expect to see under sexual selection in a given species (e.g., what *kind* of weapon, what *kind* of ornament) or even if a weapon or ornament will occur.

In this review, we consider sexual selection in a sub-order of the true bugs, the Heteroptera (Order Hemiptera). Widely distributed across habitats across the globe, from the tropics to the high Arctic tundra, with more than 40,000 described species, the Heteroptera are a common group of insects, including numerous important pest species, as well as some more beneficial, predatory natural enemies of pests. First, we will briefly introduce the Heteroptera. Then, we will focus on four sexual signaling systems found across the Heteroptera, exploring their associations with three broad habitat types, before describing a range of examples in more detail. Finally, we will consider how we might progress to a more predictive theory of sexual selection, linking ecology and patterns of sexual selection more explicitly.

## 2. The Heteroptera

The Hemiptera (or “true bugs”) is the largest of the hemimetabolous insect orders, encompassing around 7% of known insects, and includes four major sub-orders, the Sternorrhyncha, Auchenorrhyncha, Coleorrhyncha, and Heteroptera [38]. The Heteroptera is the largest of these sub-orders, representing approximately 40% of all Hemiptera, and itself consists of seven infraorders, 76 families, and approximately 42,300 species, which display a wide diversity of adaptations to a range of different lifestyles [39,40,41]. The Heteroptera occupy both terrestrial and semiaquatic habitats, including one marine dwelling genus (*Halobates* [42]), and due to their abundance, they are a key component of terrestrial and aquatic food webs [43]. They have piercing, sucking mouthparts, adapted for feeding on fluids, which enables a wide variety of diets, from plant vascular fluids to blood and other liquids [44]. They exhibit a great range of variation in morphology, reproductive behaviors, and associated adaptations.

The Heteroptera are being increasingly well-studied in terms of their ecology, life history, and behavior [45,46,47,48,49,50]. There are four main forms of sexual communication present within this sub-order: chemical communication, including sex pheromones [51,52]; stridulation [53,54,55]; substrate-borne vibration signals [4,56,57,58]; tactile communication via antennation [49,58,59]. By studying a range of sexual signals in terms of three very broad classes of habitat in which they occur, we wish to ask whether we can identify patterns, or associations, between habitat type and kind of sexual signal deployed. We will discuss these four methods of sexual signaling in terms of how they vary and how they are shaped by environmental context. For each modality, we will discuss the broad patterns uncovered in the methods, before considering a range of specific examples.

### 2.1. Chemical Signaling

As with the vast majority of insects, chemical communication is the predominant form of communication in the Heteroptera, and their chemical ecology has received a lot of attention, especially in terms of combating agricultural pests ([7,51,60,61,62]). In terms of terminology, sex pheromones are odors that convey information to the opposite sex for the purpose of mating, acting as attractants, or in courtship, whilst chemical signals more generally need not be limited to reproduction, also being used in other behaviors such as defense [7,63]. In the context of mating, chemical signaling, and sex pheromones can intersect. For instance, an aggregation pheromone, which is not a sex pheromone, may not directly influence mating, but may provide mating opportunities [7]. Adult male brown marmorated stink bugs (*Halyomorpha halys*) produce chemicals that attract males, females, and nymphs into aggregations [64]. Under experimental conditions, male and female *H. halys* were as likely to fly as each other, but females tended to fly for longer distances, perhaps associated with mate-searching via visiting aggregations [64]. Changes in internal chemistry may also trigger the production of chemicals, directly or indirectly, that are used to indicate the reproductive status of individuals [7].

From our survey across the 76 Heteroptera families, we found chemical signaling in families present in plant and leaf-litter habitats, but no records of Heteroptera in semiaquatic habitats using chemical signals. Plants can influence the production, release, enhancement, or disruption of chemical communication. For instance, in some cases, the signal cannot be produced without the host plant [65,66,67]. Although chemical signaling is recorded less often in families found in leaf-litter habitats than in plant-associated habitats, it is still prevalent as a commonly found sexual signal in this habitat type. For Heteroptera inhabiting leaf-litter, including dead leaves, moss, shed tree bark, or decaying plant matter, survival may depend on cryptic strategies. For instance, the Heteropteran infraorder Dipsocoromorpha, which contains five families of bugs, comprising about 300 species, are all leaf-litter dwelling and cryptic [49] (pp. 99–109). The production of long-range sexually signaling chemicals may better inform mates to the location of a cryptic bug whilst they remain hidden than other forms of sexual signaling, reducing the energetic costs of searching for an appropriate mate [68].

The lack of chemical signaling in semiaquatic habitats might seem unsurprising at first glance, since chemicals may be washed off, dispersed, or dissolved, interrupting the transfer of the signal, or disrupting its information content [69]. However, many aquatic and semiaquatic species, including anurans, algae, and cyanobacteria, use chemicals to communicate, indicating that, at least for these taxa, it is a viable method of communication in this habitat [69,70,71]. In addition to this, several semi-aquatic Heteropteran families use chemicals in defense, for instance, males of the giant water bug subfamily Lethocerinae use chemical defense to protect their eggs from predatory ants [72]. Therefore, it is not clear why sex pheromones appear not to be present as a form of Heteropteran sexual communication in semi-aquatic habitats.

Chemical signals must pass between the signaler and receiver. This requires not only the production of the chemical but also the receiving of, and the response to, the chemical by the receiver. Therefore, chemical composition and receptors evolve in parallel, and those used in species- and mate-recognition will be evolutionarily constrained by these functions [7]. Insect antennae often host pheromone signal receptors and have been found to coevolve with the olfactory system, developing unique and specific receptors for the optimization of signal and cue detection [73]. The evolution of antennae and their involvement in signal perception is also constrained however by a trade-off between the cost of the antennal structures and the benefit of increased pheromone detection [73,74].

Chemical signals can be both long-distance and short-range. As such, sex pheromone use can be categorized into three processes involved in mating: species recognition, mate recognition, and mate assessment [63]. Species recognition chemicals tend to be longer range, more volatile, and are beneficial in reducing the energetic cost of finding appropriate mates [68]. Mate assessment chemicals tend to show the most variation in amount produced or chemical composition, allowing the assessment of the quality of a mate, if such an assessment is favored by selection. For instance, age, fertility, and mating status may be garnered from chemical signals and used to assess a potential mate [62]. Pheromone levels released by virgin *Stenotus rubrovittatus* (Miridae) females has been found to decrease with age and to decrease when a female have been mated, when compared to virgin females [75]. This variation in pheromone production gives valuable information for males assessing the quality—such as mating status—of a mate.

Pheromones may also be produced, released, enhanced, or disrupted in response to host plant volatiles, and therefore plants can play a vital role in mediating insect sexual communications [66]. Broad bean plants, *Vicia faba*, have been found to release volatiles that attract the European tarnished bug, *Lygus rugulipennis*. In addition to this, the sex pheromones released by mated female *L. rugulipennis* were enhanced when females were active on the broad bean plant, indicating that the plant is involved in aggregation and female-male pheromone communication [76]. This enhancement of chemical signaling due to the host plant species is also demonstrated by *Narnia femorata* males (Coreidae). If they develop on cactus fruits, males will produce a more enticing odor than those that develop without access to fruits. The improved odor as a consequence of their diet increases their mating success [77].

The chemical signaling of bugs is often exploited to produce pheromone lures that target herbivorous, crop-damaging pest bugs [65,67,78]. Chemicals are also used in defense, commonly paired with bright colors as an aposematic deterrent for predators, for instance in seed bugs (Lygaeidae [46]). These chemicals are often sequestered from poisonous plants that are part of the diet; for example, the lygaeid *Oncopeltus fasciatus* sequesters cardenolides from milkweed plants as a defense against predation [79]. The extent to which the defensive roles of such chemicals, shaped by natural selection, interact with possible secondary sexual function, remains a field ripe for broader exploration, in both the Heteroptera and more generally.

### 2.2. Stridulation

Stridulation involves one part of the body being rubbed against another to produce sound, and typically this involves a specialized stridulatory organ consisting of a so-called “scraper” and a “file”, with the sound produced as the scraper is dragged across the file [80]. Low-frequency signals are produced by the movement of the abdominal tergal plate (previously termed the tymbal), which for many of the Heteroptera are morphologically distinct [53]. Most Heteroptera that produce sound does so via stridulation, although other mechanisms include abdomen vibration and hitting a substrate (see below [81]). Stridulation in heteropteran bugs has been found to occur in both sexes, and from nymphal through to adult stages, although nymphs presumably only produce stridulation to repel predators [8,82]. Stridulatory signals are both auditory and vibrational, the latter often detected using the antennae to follow the signal to its source [53].

Sound production via stridulation is energetically costly for Heteroptera due to their small size, and most sounds produced this way are of a high frequency which are subject to degradation across space [8]. The efficacy of the stridulatory signal produced may thus be particularly influenced by the environment in which it is produced. In terms of the two potential components of stridulation signals, the frequencies of the sound produced can extend up to 10 kHz, but vibrations have a lower frequency which peaks at 100 Hz, which when produced at the same time manifest as broadband signals [53].

Stridulation was the most common form of sexual communication in all habitat types. Sounds produced by stridulation are mostly high frequency and travel further and are less distorted when traveling in open environments, when compared to environments with physical barriers [8]. Bugs found in plant environments are elevated from the ground and in more open environments, allowing stridulation to act as a longer-range sexual signal, attracting mates from further afield [83]. Plants are the most widely used substrate for transmitting vibration signals, produced by both stridulation and abdominal vibration, transmitting vibrations more successfully than other substrates such as leaf litter [84,85].

Semi-aquatic environments differ from terrestrial environments in terms of temperature, pressure, and density, all of which will affect how signals travel. Water density causes the sound to travel faster, but in a lower and narrower band peak frequency, dampening especially loud sounds, as found to occur in signals produced by the water boatmen species *Corixa dentipes* and *C. punctata* (Family Corixidae [53,55,86,87]). These pulses are emitted at a relatively narrow band peak frequency of 1.5–2.8 kHz, much reduced from the 10kHz peak in a terrestrial environment [53,55]. Due to these limitations, sound production in semi-aquatic insects is highly specialized [82]. For example, when diving, corixids are contained in a respiratory air bubble, the volume of which affects the sound produced by stridulation. These submerged respiratory air bubbles play an important role in communication, as the stridulatory sound of one animal will induce resonant oscillation in the air bubble of a nearby animal, acting as a sound radiator, thus providing a very different application of stridulation in the role of transmitting signals involved in sexual communication [55]. It has been theorized that females may use these underwater oscillations as a non-visual method to locate males, beneficial for a species that largely copulates in the dark [55]. As such, both water bugs (*Micronecta*) and water boatmen (also in the Corixidae) have been recorded as producing sounds underwater via stridulation, using trapped air reserves to produce and receive signals [55,82,88,89,90,91]. In shallow water, sounds may be reflected from the water/air interface or be lost in the sound of the water, another issue that may distort the signal and the information contained [82]. However, stridulation clearly performs well enough from the water’s surface, as it is a loud form of communication with peak frequencies extending up to 10 kHz in some cases, loud enough to be heard over the noise of moving water [53].

Importantly, different stridulatory songs may be produced by the same individuals throughout the mating cycle, to communicate location, choice, acceptance, or rejection, depending on context. For instance, species in the stink bug (Pentatomidae) and assassin bug (Reduviidae) families produce different songs for acceptance, rejection, and rivalry during courtship [92,93]. Specifically, females of the assassin bug *Triatoma infestans* produce a long sequence of repetitive syllables to deter eager to mate males, and possibly also to deter predators [92]. *Rhodnius prolixus* females (assassin bugs) will also produce a string of repetitive syllables to reject male attempts at copulation, a method that can be up to 100% effective [94]. As well as producing different stridulatory sounds, these different signals can induce different behavioral responses among different receivers. Within *Cenocorixa*, a small genus within the Corixidae males, and females emit sex-specific stridulations. The males produce an antagonistic signal which spaces out calling males, whilst also attracting females and facilitating pair formation. Females respond by remaining stationary and stridulating, indicating a soliciting behavior in contrast to the male’s antagonistic behavior [54].

Sound production can be dangerous of course, risking advertising your presence to other organisms, including predators and parasites. Indeed, predation is probably one of the key constraints on sexual signaling across environments. This is in part due to the narrow separation in audible frequencies that can be produced and received within a given habitat [8]. Some Heteroptera call from aggregated groups, which decreases some of the costs imposed by predation and parasitism, via the dilution effect. However, as the desired effect of these signals is to attract conspecifics, it is difficult to identify if the decrease in predation risk afforded by large groups is evolved or coincidental [8,95]. It has been suggested that broad headed bugs, *Alydus pilosulus*, stridulate from within aggregations in order to isolate females from the accumulation of bugs before mating, using the safety of the crowd to avoid predation [96].

The effective transmission of stridulatory signals may also be compromised by interference and overlap of signals from other insects in the environment. These hetero- or con-specific interactions will in general be unintentional and have only negative consequences for the signalers and receivers involved, decreasing the effectiveness of mate location and recognition [4,97,98]. The stink bugs *Eushistus heros* emit narrowband, low frequency stridulatory signals and communicate via sex specific vibratory signals, sung in duets during calling, courtship, and rivalry. When two individuals call at the same time, the overlap changes the amplitude pattern of the call. When this occurs, males and females will change the time parameters of their calls in order to increase the frequency difference between the signals and avoid confusion [99]. Calling within a narrow frequency window presents a trade-off, reducing the risk of signal interference but also restricting the distance that the signal can travel. This results in a mosaic of species-specific stridulatory signals each pushing against each other in signal space in an effort to maximize transmission success [4,22,97].

The increased occurrence of stridulating Heteroptera in leaf-litter habitats compared to those that communicate via abdominal vibrations suggests that the sound produced by stridulation may communicate better than vibration signals in this habitat. Leaf-litter has a dampening effect on vibrations, restricting the distance that they can travel (see abdominal vibration below [100,101,102]). One limitation of the Heteroptera in sound production is their small size, causing stridulation to be energetically costly, producing sounds of a high frequency which quickly degrade through space [8]. This issue may increase exponentially for cryptic species dwelling in leaf litter, which are often smaller in size than heteropterans found in plant or semi-aquatic habitats, resulting in this form of communication perhaps being energetically detrimental and therefore less common [45,47,49,50].

### 2.3. Abdominal Vibration

Abdominal vibration is most commonly associated with plant habitats in the Heteroptera. Abdominal vibrations are typically a short-range method of communication and are considered to be safer from eavesdroppers [22,103]. In a plant habitat, vibrations may be easier to locate than airborne or visual signals, which may be blocked by leaves, stems, or other structural components of the vegetation or canopy [22,103]. Abdominal vibrations are also used by families in semiaquatic habitats, although stridulation was more common (see above). Surface vibrations in semiaquatic habitats can be used to identify and differentiate between predators, prey, hetero- or con-specific male and females [104,105,106]. Abdominal vibrations are not commonly associated with leaf-litter environments, likely due to the dampening effect of leaf-litter which prevents the signal from traveling over any real distance, as demonstrated in research regarding wolf spiders and the transmission of vibration signals [100,101,102].

Vibrational signals travel along the substrate that the individual is perched on, causing this to be typically a shorter-range signal of attraction [107]. This method is more commonly found in smaller and more weakly sclerotized species, which may lack the ability to form sound-producing organs due to their soft bodies and who incur a high cost of sound production due to their small size (see stridulation [107]). Vibrational signals are produced through rapid tremulation or percussive vibration of a body part, usually the abdomen or the wings. Tremulation consists of moving the substrate by the release of energy by the individual thereby creating waves, however, percussion involves actual contact with the substrate, beating it with the vibrating organ [107]. Tremulation results in a more constrained and consistent signal and is more common within the Hemiptera than percussion, and communication via vibration itself is considered to be ancestral for the hemipteran order, occurring in the five monophyletic lineages (Sternorrhyncha, Fulgoromorpha, Cicadomorpha, Coleorhyncha, and Heteroptera; [107,108]).

Abdominal vibrations are low frequency, peaking at 90–140 Hz, and are, therefore, involved in short-range communication However, they can also be paired with other forms of communication to produce a broadband signal [53,58]. The distance that this signal can travel is also dependent on the substrate through which the vibration travels, as dampening in the path can decay the signal [109]. Vibration signals have an attenuation of up to 20 dB when the signal transmits from leaves to stalks or stems, but this can vary, for instance, if the stem is green versus woody [110]. Harlequin bugs, *Murgantia histrionica*, often inhabit *Brassica oleracea* plants (cabbage, broccoli, kale, etc.) which are characterized as having compact heads with layers of large leaves with rod-like stems and veins. These veins carry vibration signals more efficiently than the head of the plant [111], and harlequin bugs are able to estimate the distance of the signaler based on the differences in the peak amplitude of the song at different locations [111].

Signals that travel through substrates have several advantages; due to the limited and targeted range of vibrational signals, they are often regarded as “safer” with respect to eavesdroppers, but when occupying plants, it may also be easier to identify and locate these vibrations than visual and airborne signals, which may be obscured by the plants [22,103]. For instance, the parasitoid wasp *Telenomus podisi* eavesdrops on the communication of stink bugs, following female vibratory signals along plant stems to the location of her eggs [112]. Both intended and unintended receivers of vibration signals recognize them via sensillae, which can be located externally, for instance, the campaniform sensilla and other mechanoreceptors located in the cuticle, or internally, via scolopidial sensilla organs, specifically the subgenual organ, joint chordotonal organs located in the legs, and the Johnston’s organ located in the antenna [53,113]. For example, male Southern green stink bugs, *Nezara viridula*, follow the songs of females along the branches of plants. When they reach a junction, males will compare the vibrations of two branches with their legs and antennae, employing vibrational directionality to locate the source of the signal [103,114].

Vibrational songs vary greatly in their frequency, amplitude, and behavioral response. Many species have sex-specific abdominal vibration songs. For instance, only male spined soldier bugs, *Podisus maculiventris*, produce vibratory signals, consisting of abdominal vibrations, percussion of the front legs, and tremulation of the body. Males release pulse trains of vibrations which induce searching behavior in females [58]. More commonly, males and females will both produce species-specific vibration songs. Female neotropical stink bug songs contain pulses that differ between species in duration and repetition rate, to which males respond with courtship songs that are likewise species-specific in temporal structure and amplitude [115]. The green neotropical stink bugs *Chinavia impicticornis* and *C. ubica* both live and breed on the same plants and sexually communicate via abdominal vibrational signals. But these two species produce calls differing in their spectral and temporal characteristics, reducing reproductive interference, and preventing the initiation of hetero-specific courtship behaviors, resulting in species isolation [116] (see Stridulation). However, species-specific abdominal vibration calls are not always easy to differentiate. Stink bugs *Acrosternum impicticorne* and *Euschistus heros* [115] have very different calls, but they still occur within the same frequency range and, therefore, could still provide difficulties when differentiating between them [115]. When male green stink bugs, *Nezara viridula*, were presented with the songs of a conspecific female and a heterospecific female, males made several orientation errors in response to the overlapping signals, and the majority located the heterospecific source. This confusion due to overlapping signals may cost mating opportunities and reveal the limitations of vibrational communication [117].

Vibration communication can be used in mate evaluation or used to induce sex-specific behaviors. Within the family Pentatomidae, males and females often have multiple songs that are sung in sequence, guiding the opposite sex through mate location, courtship, and copulation [111,115,118,119,120,121,122]. A study of three stink bug species (*Chlorochroa uhleri, C. ligata*, and *C. sayi*) determined that all the species started with a calling song, followed by a courtship song, and finally a copulatory song, but that only two of the species (*C. ligata* and *C. sayi*) produced an additional song in the event of interactions with rivals [123]. The sequence of songs produced can be complex. For example, male harlequin bugs produce five different types of songs, in comparison to the female’s one song. Four of these songs take place before copulation, proceeding through several ascending harmonic frequencies and consisting of pulse trains of different durations and repetitions, leading through to copulation itself. The final song was a rival song produced when two males compete over a female [111].

It is also common to find vibrational signals produced in aquatic environments; vibrations produce surface waves and ripples which send out signals to rivals, the opposite sex, and also predators [104,105,106]. Water striders, for example, have a specialized sense organ, the trichobotria, which can distinguish between ripple frequencies in the water and are used to differentiate between the threat of an oncoming competitor or predator [104]. This ripple communication (and eavesdropping) can also distinguish the sex of the approaching individual and between predators and prey [104,124]. *Limnoporus dissortis* and *Limnoporus notabilis* males will produce a repel signal when encountering a vibration-induced wave, but females do not, unwittingly identifying themselves as a potential mate [106]. Wilcox & Stefano [125] also showed that male vibrational signals deterred other males, and so contribute to the success of male mate-guarding (and as mate guarding also improves female foraging, by reducing further male harassment, these signals also improve female foraging success) [125].

Signaling via water vibrations may also be used in coercive mating. Male Asian water striders, *Gerris gracilicornis*, once mounted will intimidate females into mating by using water vibrations to attract predators if she does not accept the copulation attempt. This strategy aims to lower the fitness of the female and is thought to have evolved as part of a coevolutionary arms race in response to the female’s morphological shield that protects her genitalia from coercive intromission [126]. This tactic was found to be employed more by large males, since smaller males were unable to produce sufficiently strong ripples to be able to attract the notice of predators, due to their shorter legs not reaching the water once mounted, and so these smaller males more often used non-signaling courtship methods [127]. Therefore, in this instance, this tactical and coercive use of vibrational communication is effective only as a size-dependent reproductive tactic.

### 2.4. Antennation

The final form of sexual signaling we will consider is antennation. Antennation refers to the antennal grooming of the partner during courtship and mating. Darwin [128] proposed that sensory organs play an important role in sexual selection, referring to the locating, identification, and attraction of a mate, a suggestion that was, however, largely ignored at the time [73,129,130]. Within the Heteroptera, antennae are involved in a variety of sexual behaviors, from the detection of chemical odors used in chemical signaling, the localization of vibratory signals, courtship grooming, all the way through to physical restraint during copulation [73,114,122,131]. This has resulted in a range of complex morphologies of antennae, shaped by the specific pressures placed upon them by sexual communication over long or short distances and sexual conflict over mating. For instance, antennae involved in receiving chemical signals are larger and often filamentous, as larger antennae contain greater numbers of the sensillae involved in the detection of sex pheromones [73].

Once in close proximity to the source of a sexual signal, the antennae may then be engaged in short-range sexual signaling. A remarkable nymph of the bug *Magnusantenna wuae* was recently discovered in a sample of Burmese amber, displaying large and intricate leaf-like antennae. Although suggested to play the role of a delicate sensory organ involved in sexual selection [129], such structures in a sub-adult nymph may also be the makings of an even more impressive visual display in the adult bug. Similar leaf-like appendages are, of course, well-known in the leaf-footed bugs (Coreidae [132]).

Once a mate has been identified and approached, antennation can involve tapping, antennal vibrations, and body-surface palpitation with one or both antennae in order to communicate with and assess the mate [133]. Assessment includes the willingness of the mate to copulate, and also the potential quality of the mate, for instance via their size or cuticular pheromones [134]. *Lygus hesperus* (Miridae) males, when presented with a recently mated female, may antennate her abdomen and then move away without attempting to mate, perhaps in response to her mating status [135]. It has been suggested that the female spermatheca in *Lygus* species contains volatile compounds that advertise the female’s fertility, a signal that would be curtailed after mating [60,136].

When identifying the Heteroptera families that use antennation in sexual communication, we accredited only those that use it as a signal, not as a method of receiving signals (the latter being an obvious component of many instances of the other three modes described above). Antennation-as-signal was more commonly associated with families found in plant habitats, was the least common form of sexual communication in leaf-litter habitats, and was not found at all in semi-aquatic habitats. Antennation is a short-range form of communication, often used just prior to, during, or after copulation, and often employed in courtship to stimulate and maintain mating or to distract from other potential mates [122]. Therefore, this method of sexual communication may be more important in plant environments, where potential mates are more visible and therefore more vulnerable to competing males. In terms of species living in leaf-litter habitats, as these bugs are often cryptic and therefore concealed when mating, they may be less likely to be subject to intense competition from rivals. However, it is also possible that this method of communication has not been seen in these families because of under-reporting (since detailed observations of matings are required).

Antennation has also been explored as a behavior present in cryptic mate choice. Cryptic mate choice refers to selection that occurs during copulation, and either before, during, or after insemination, and so is typically hidden from direct observation. This can occur when essential information leading to mate choice can only be gathered once mating has begun [137,138,139]. For instance, male leaf-footed bugs, *Leptoglossus clypealis*, will gently antennate the female’s head, antennae, and abdomen with his antennae and front legs during courtship. In response to this, the female becomes quiescent, opening her genital chamber and allowing the male to mate. Females may require several bouts of this courtship before they are prepared to accept a male [122]. Similar behavior is also present in some stink bug species, for instance, *Thyanta pallidovirens* [122]. Antennation has been described twice in Lygaeidae species (*Nysius huttoni* [140] and *Oncopeltus fasciatus* [141]), although species within this sub-family are more widely known for their lack of courtship and high rates of mating failure, which itself may be the result of sexually selected cryptic choice [142,143].

Antennae sometimes also play a more active role in mating, with male antennal morphology adapted to grabbing and holding females during copulation. This is the case for *Rheumatobates rileyi* males, whose elaborate antenna grasp resistant females during copulation. The antennae of these water-striders are wrench shaped with a “spike”, formed of several bristles, which fits into the groove that runs between the female’s thoracic segment, eye and head capsule, a pad which sits beneath the female’s eye, and a hook which grips the female, and which are important in allowing the male to leaver his body on top of the females [131]. Those individuals that have a reduction in grasping traits have reduced mating success, demonstrating the direct role that antennae can play in sexual selection [131]. *Harpocera thoracica* (Miridae) also have specialized antennae involved in grasping females, and in this case, the antennae have adhesive setae which grab females during mating [144]. Importantly, while this use of the antennae is not a form of sexual signaling, it does actively prevent antennae from being used in this way, a loss of one function to accommodate another. As such, the co-option of antennae into sexual struggles of mating provides a very different kind of example of how the environment may constrain opportunities for sexual signaling; in this case, how the environment shapes the mating system and the kind of mate competition that emerges, shaping selection on a primarily sensory organ to take on a new, sexually selected function.

### 2.5. Methods

Data about habitat use and sexual signaling were collected for 76 Heteroptera families (Appendix A).

Habitats occupied were classified as (i) ground-/leaf-litter living, (ii) plant-associated, and (iii) semi-aquatic. If recorded in more than one of these habitats, then all habitat types were included for each family. Ground living includes species primarily associated with life on soil or rocky substrates, or in the leaf litter (including dead leaves, moss, shed tree bark, or decaying plant matter). Plant-associated living includes any being primarily found on trees, vines, leaf surfaces, or grasses and shrubs. Semiaquatic refers to any or all adult use of aquatic habitats, be they marine (for instance, *Halobates*, the sea skaters) or freshwater [49,93,145].

Data were collected about the presence of four types of sexual signaling: (i) stridulation, (ii) vibration signaling (via abdominal vibration), (iii) chemical signals, including the production of pheromones, and (iv) tactile signaling via antennation. 

Families were scored for presence (1) or absence (0) for signaling mode and habitat occupation; families were sometimes found to exhibit multiple signaling modes or be present in multiple habitats, in which case they were scored for presence in each. The data are presented in Table 1 (references for data points can be found in Appendix A). There is significant heterogeneity in the occurrence of signaling mode across habitat types (χ^2^_6_ = 16.82, *p* < 0.01), suggesting that habitat helps influence the mode of sexual signaling across Heteroptera families. Thirty heteropteran families were not recorded as using any of the four signaling modalities identified, whilst 29 families were found to use one modality, 11 families used two modalities, and six families used three (Table 2).

## 3. Discussion

Communication is key in competition for mates and gametes, and therefore how the Heteroptera sexually communicate plays an important role in sexual selection in these insects. Here we have reviewed four prominent methods of sexual communication displayed by the Heteroptera, providing examples to highlight the variation found across the sub-order. From these examples, clearly, the environmental context in which sexual competition takes place can influence the success of any given competitive strategy or behavior. To what extent then do the habitats of heteropteran bugs shape how sexual selection plays out in these insects? When we consider how our four modes of signaling are distributed across three, admittedly very broad, habitat types in the Heteroptera, we find significant heterogeneity in how these signaling modes are deployed. For instance, stridulation was associated with families found in plant and semiaquatic habitats, and antennation was associated with families found in plant habitats. How might we go forward from such broad-scale approaches?

To begin with, we note that our analysis is very much prospective, and the data presented have a number of limitations, most notably in terms of the collapsing of data to the family level, the inherent study biases of certain groups and phenotypes, as well as the under-reporting of Heteropteran signals more. Moreover, classifying environments is a complex challenge in itself, let alone associating those environments with patterns of sexual selection. Nonetheless, some promising patterns emerge and prospective analyses such as these are needed if we are to move towards a more comparative study of sexual selection, and perhaps build a more predictive framework. As such, the aim of our study was not to necessarily point out novel relationships between communication and environment, but rather to highlight that patterns in sexual communication can be found in terms of the environment in which they occur. Whilst we certainly need more formal, phylogenetically-controlled comparative analyses to interrogate more robustly associations between the environment and what kinds of traits become sexually selected, we hope that our review will encourage both the continued study of a diverse range of organisms and scrutiny of the existing literature for researchers’ taxa of choice.

### Predicting through Patterns

While sexual selection may be quite simple in outline—namely the selection that arises from the competition for access to gametes [146]—the ways in which that competition can manifest itself are extraordinarily diverse, leading to the great diversity of sexual traits that we see [1,6,128]. However, the mechanisms of competition for mates do not act in isolation. They are constantly shaped by the environment that surrounds them [15,17,18,147,148]. There are, however, two ways that we can view the role of the environment. First, we can consider the environment as providing a range of constraints, physical or biotic, to successful signal propagation. This is perhaps often the perspective taken when addressing how physical aspects of the environment influence signal design and efficacy. Second, we can instead consider how different environments provide different opportunities for signals to evolve and be utilized. For instance, life in aquatic environments will involve new ways of finding food (e.g., evolving to respond to ripples and water movements made by prey) or indeed living underwater (e.g., developing structures to allow or enhance a plastron to form around the body, prolonging diving time). These adaptations may then facilitate the evolution of vibrational signals in the former case, or the amplification of stridulatory signals in the latter.

Our review suggests that for insects that rely on long-range forms of signaling for finding mates, chemical signals are likely to dominate in terrestrial habitats, although the extent to which such signals are themselves under sexual selection, or rather set the stage for competitive mate searching in the other sex, still remains poorly understood. For chemical ecologists, this is not a new finding, of course. Among close-range signals, that are perhaps more likely to be directly influenced by sexual selection, chemical signals are still common, but other sexual signaling modes also become more important. Here, for instance, insects living on plants are more likely to make use of vibrational signaling, and the specific properties of water in aquatic habitats likewise favor vibrational signals. More cryptic habitats may also favor more cryptic forms of signaling, although uncovering such signals will remain challenging, and despite scrutinizing several hundred papers for data on 76 families, our dataset is still extremely limited in its level of detail.

The idea of searching for patterns among sexually selected traits in order to attempt predictions of what kinds of traits might evolve under what circumstances has been discussed before. Andersson & Iwasa [149] published a literature review which, amongst a review of the mechanisms of sexual selection, discussed the patterns present in sexual traits that were found by the review studies of Møller & Pomiankowski [150] and Ryan & Keddy-Hector [151]. In particular, in their review of birds, reptiles, amphibians, and fishes, Ryan & Keddy-Hector [151] found that females prefer traits that are found in greater quantity and therefore elicit greater stimulation, for instance, tungara frog (*Engystomops pustulosus*) males are more attractive when they include a greater number of “chucks” in their song [151]. What individuals of the opposite sex find attractive about the calls and displays of the other sex has, of course, long been of interest, not least in terms of trying to understand why such preferences evolve and discriminating between different theories for that evolution (such as good genes, the Fisher process, or sensory bias [1,31]). Our focus here though is more about why certain kinds of signals begin any such process. And as discussed above, John Endler has pioneered discussions of how environmental conditions influence signaling behavior, linking sensory systems, signals, signaling behavior, and habitat choice in the evolution of the diverse range of signals that we see (e.g., Endler [5]).

Our work suggests that physical constraints within different habitat types will influence what sexual signals are commonly used to communicate, most clearly shown here at the very basic level of terrestrial versus semi-aquatic. Different habitats, with different vegetation structures, will influence how stridulatory or vibrational acoustic signals are propagated, as well as the extent to which species live cryptically, in leaf litter or similarly unobtrusive niches. Our study is in support of the acoustic adaptation hypothesis (AAH), which posits that animals should use signals that travel well within the environment that they are found [152,153,154,155,156]. A review of playback tests in birds, mammals, insects, and anurans revealed that transmission properties consistently varied by habitat (e.g., closed habitats degraded signals more than open), confirming that habitats should provide different selection pressures on acoustic transmission [154]. They found that 63% of studies demonstrated at least some evidence of species-specific acoustic fidelity, which confirmed the results of other similar reviews [154]. Hardt & Benedict [154] outline many reasons why acoustic fidelity may not be present, however, including if calls are innate or learned, variation in habitat fidelity, species-specific responses to evolutionary pressures, and also inconsistencies in methodologies [154]. Although not present in all cases, there is a definite habitat-specific selection of acoustic signaling across multiple taxa. And changes in signals due to changes in habitat, for instance, through urbanization, are becoming more widely researched and provide more examples of how selection pressures can result in sexual signaling that compensates for the surrounding environment. Anurans, in particular, have demonstrated a change in call rate or call complexity in response to novel selection pressures introduced by an urban environment [157,158,159,160,161].

In terms of the biotic environment, plants are not just structural components of the environment. Instead, as we have seen, they can indirectly and directly influence sexual signaling in terms of the interaction between insect chemical signals and the volatile chemistry of the plants themselves, or indeed the chemical constituents of the plants, for instance in terms of plant secondary compounds. The host plant may influence the quality, quantity, and timing of the release of signals and, therefore, when a plant is essential for the production of a signal, the distribution of the species is tied to the host plant [66,162]. This connection affects the distribution, expansion, and sometimes even survival of a given species. Changes to host plants can change the cuticular hydrocarbon (CHC) profile of a species or its pheromones, and if they are unable to adapt in time, they may be unrecognizable and therefore fail to mate [162]. For instance, mustard leaf beetles (*Phaedon cochleariae*) prefer to mate with individuals reared on the same plant as them, as opposed to those reared on different plants, due to the composition of their CHCs [163]. Therefore, the biotic environment can play a large role in what methods of sexual communication are present, including chemical signaling.

The Heteroptera also provides some compelling examples of how predators and parasites can likewise shape sexual signals, including in terms of the intimidation tactics used by male Asian water striders to “persuade” females to mate by producing ripples that attract predators if they resist (see above). Indeed, the constraints imposed by predators and parasites may well be crucial in terms of delineating what kinds of traits are sexually selected—being visually or acoustically showy may be exceptionally costly. This biotic component of the environment may well explain why many organisms do not express elaborate secondary sexual characters, and why post-copulatory sexual selection may be more prevalent than pre-copulatory sexual selection in many species (as is likely the case in the Heteroptera [132]).

While the physical properties of the environment, and the potential threats from predators and parasites eager to take advantage of the need for animals to signal for sex, will all influence what kinds of signals evolve, perhaps the most important aspect of the environment in shaping sexual signals is the opposite sex. As we have described, in some cases—such as duetting brown stink bugs (*Eushistus heros*) which duet through the calling, courtship, and rivalry phases of mating—males and females very intimately shape the signals used in communication [99]. The various sensory biases and mechanical opportunities for attracting and/or coercing the opposite sex clearly shape how both males and females produce and receive sexual communications. This aspect of the ecology of sexual interactions is perhaps the best studied; the challenge is now to put our often rather detailed understanding of female-male interactions back into its broader ecological context, to try and identify commonalities in how different habitats shape the evolutionary trajectories of sexually selected traits. Within a rapidly changing environment, this could be essential to predicting how species will persist and survive, as selection changes in environments which we ourselves are steadily influencing and interfering with.

## Figures and Tables

**Table 1 insects-12-01079-t001:** The number of heteropteran families found in one of 3 habitats: leaf litter, plant, or semi-aquatic habitats, and use one of 4 methods of sexual communications: stridulation, abdominal vibration, chemicals or antennation.

Names	Abdominal Vibration	Stridulation	Chemical	Antennation
Semiaquatic	4	15	0	0
Plants	8	16	10	10
Leaf-litter/floor	1	5	5	2

**Table 2 insects-12-01079-t002:** The number of heteropteran families found to exhibit 0, 1, 2 or 3 signal types and found in 0, 1, 2 or 3 habitat types. Lack of reporting and limited study of many families will influence current estimates of signal and habitat diversity across the Heteroptera.

		Number of Signal Types			
		0	1	2	3
**Number of habitats**	**0**	7	2	1	0
	**1**	19	21	10	5
	**2**	3	4	0	1
	**3**	1	2	0	0

## Data Availability

All data can be found in Appendix A.

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
