# Peer review of "Environmental Correlates of Sexual Signaling in the Heteroptera: A Prospective Study"

_insects, 2021, doi:10.3390/insects12121079_

Round 1

Reviewer 1 Report

I think the authors did an adequate job of addressing my previous concerns.

Author Response

Dear reviewer 1, 

Thank you very much for the time you took to review our paper. We are glad that you believe that we have answered your comments to a satisfactory standard. The comments that you provided were a great help in improving our work. 

Thank you, 

Eleanor Gourevitch 

Reviewer 2 Report

This review examines the signalling modes of the Heteroptera and attempts to links them with the environment they inhabit. The manuscript is interesting and it's results intriguing, if somewhat preliminary due to the availability of data. Overall I think it will make a good contribution to the literature on this subject and have only one minor suggestion detailed below. 

I am sad to see that Figure 1 has been removed. I always think it is nice to have figures like this to help readers not familiar with the taxa picture the species being described. My opinion would be it could be added back in, but with a small caveat that the common names used in the figure legend should match those used in the table in the supplementary materials, to aid in locating the corresponding references. 

Author Response

Dear reviewer 2, 

Thank you for your comments. We recognise that figures can be interesting and informative additions to papers however we decided to remove this figure based on the comments of reviewer 1 and we have decided to stay with this decision.

We appreciate the time you have taken to review our paper. 

Best wishes, 

Eleanor Gourevitch 

Round 2

Reviewer 2 Report

I'm happy  to recommend this manuscript for publication.

This manuscript is a resubmission of an earlier submission. The following is a list of the peer review reports and author responses from that submission.

Round 1

Reviewer 1 Report

Main comments:

This study aims to understand why sexual selection leads to the evolution of different types of traits in different species. As a first step towards a more predictive theory of sexual selection, the authors review sexual signaling in Heteroptera, a sub-order of Hemiptera. They evaluate whether sexual signaling mode (chemical, stridulation, substrate, or antennation) was correlated with different habitat types and find some evidence that terrestrial versus semi-aquatic taxa use different modes of communication.

I was intrigued by the abstract and really wanted to like this paper. I agree with the authors that it would be very valuable for sexual selection studies to conduct large comparative studies to develop a framework for understanding broad-scale patterns in why different taxa exhibit different types of sexually selected traits. Unfortunately, I felt that this paper fell short. The results from the review are summarized in Table 1, but the categories are too broad to be informative. The only real pattern is that semi-aquatic families do not use chemical signaling or antennation. For the other two habitats, all signaling modes are used and at similar frequencies (11-21% for plant families and 1-7% for floor families). Taken together, I do not think this review helps us understand the environmental factors that explain broad-scale patterns in signaling modes.

Another issue that is not addressed by the table is whether families use just one type of signaling or multiple types of signaling. For example, the plant families (as a whole) appear to exhibit all four types of signaling, but it is unclear whether this is also true on an individual family level, or if each family typically specializes on just one type of signaling. I think it would be very interesting to examine how many times multi-modal signaling has evolved by plotting the types of signaling used for each family on a phylogeny.

I realize that a robust, comparative analysis may be beyond the scope of this paper. I also realize that using very broad categories for habitat type and communication mode is not ideal but may be a necessary place to start. Nevertheless, without more detail, I’m not sure the quantitative comparisons added anything. In fact, the results might turn readers away from doing any type of broad-scale comparisons because they’ll see that it’s a lot of work but isn’t informative. I therefore suggest either expanding the results of the table (to at least examine the use of multiple signals on an individual family level) or scrapping the table and developing the discussion (e.g., to include concrete recommendations on how researchers should investigate these broad-scale patterns in their own systems).

Specific comments:

The authors should decide on simple names for the four signaling modes and then use the same terms throughout. For example, in section 2.3 the authors use both “abdominal vibration” and “vibrational signals”. I assume these are the same, but it would be better to be consistent with the terminology.

I am also unsure whether antennation can be classified as a signaling mode, like chemical signaling, stridulation, abdominal vibrations. Is there evidence that females act as receivers and change their behavior based on differences in male antennation? Antennation seems like a classic form of copulatory courtship, which may allow males to persuade females into mating for longer periods of time. I’m not sure if this qualifies as short-range sexual signaling. At the very least, the range and conditions over which antennation can be used seem much more restrictive than the other signaling modes, and this should be discussed.

I found the order of the sections confusing. Why do the methods come so late? Throughout the review section, I found myself asking how taxa were classified. I think the methods should come before the review of signals. I am also curious how the authors handled poorly studied taxa. Specifically, how did they decide when a family has a signaling absence (0), or if that signaling mode simply hasn’t been studied yet? For example, it is possible that semi-aquatic families do indeed use chemical signaling, but human observers haven’t been able to detect it yet. I think this possibility needs to be discussed.

I think the photos in figures 1 and 2 are nice, but they don’t do a good job of showing the different types of signaling. For example, it is confusing that E is an example of antennation when the insect isn’t even in copula. It would be much more informative to show a photo of that species performing antennation. Similarly for figure 2, it would be helpful if the authors indicated where the stridulatory organs are located.

I’m not convinced that figures 3-5 are necessary.

Line 44: Ritualized displays do not avoid actual physical contests. They allow individuals to avoid actual physical contests. Please add “allow individuals to”.

Line 81-83: The “rehabilitation of mate choice” has been a much more intense focus of sexual selection research than post-copulatory sexual selection. I would therefore flip the order of these two categories.

Line 87. You can delete “perhaps”. We are lacking a framework.

Line 89-91: I do not think we are even successful at predicting when a given species will have any kind of weapon versus any type of ornament.

Line 113-114: Perhaps it would be helpful to highlight some of this variation in a figure? (With the exception of B, the species in Figure 1 do not look incredibly diverse.)

Lines 127-208: I recommend renaming this section as just “chemical signaling” so a single term can be used to encompass both long-range signals (e.g., aggregation pheromones) and short-range signals (e.g., sex pheromones?). I also think it would be helpful if the authors defined sex pheromones so readers understand when chemical signaling and sex pheromones should (or shouldn’t) intersect. It seems to me that sex pheromones are simply a subset of chemical signaling.

Lines 140-141: As noted above, it is possible that semi-aquatic families do use chemical signaling, but it just hasn’t been detected by human observers yet.

Line 172: I suggest changing “intimately close” to “short-range”.

Line 179: What’s a semiochemical?

Lines 231-234: Seems like this sentence might fit better in the abdominal vibration section.

Methods: How did you choose the 76 families? Is that all of them, or just some of them?

I’m confused by the classification of habitats. If recorded in more than one habitat, it was scored as having all three habitats? Was it ever possible to just see two? How often did a family have multiple habitats? Are the families that have multiple habitats the same ones that use multiple signaling modes?

Table 1: What is reported in the parentheses? Overall proportions based on all 76 families? How did you deal with families that are counted in more than one habitat (as noted above)? It might be more informative to report proportions based on a given habitat or given signaling mode. (But as noted in my general comments, I find this table largely uninformative and think the authors should consider removing it.)

Lines 624-638: These factors are unlikely to play a role in the efficacy of antennation, which highlights why antennation may need to be treated as separately (see comment above).

Lines 644-648: This is where the presence or absence of antennation may be informative. If we consider antennation as a form of copulatory courtship, it may be interesting to explore whether there are environmental factors that predict when copulatory courtship is present/absent. For example, copulatory courtship may only be beneficial if pre-mating courtship/signaling is limited.

Reviewer 2 Report

All my comments are attached in the file.
